# Impact of Crystallization on the Development of Statistical Self-Bonding Strength at Initially Amorphous Polymer–Polymer Interfaces

**DOI:** 10.3390/polym14214519

**Published:** 2022-10-25

**Authors:** Yuri M. Boiko

**Affiliations:** Laboratory of Physics of Strength, Ioffe Institute, 26 Politekhnicheskaya Str., 194021 St. Petersburg, Russia; yuri.boiko@mail.ioffe.ru

**Keywords:** poly(ethylene terephthalate), polymer–polymer interfaces, self-bonding, adhesion, lap shear strength, Weibull’s distribution, Gaussian distribution

## Abstract

To investigate the mechanisms of the adhesion (self-bonding) strength (*σ*) development during the early stages of self-healing of polymer–polymer interfaces and fracture thereof, it is useful to operate not only with the average *σ* value but with the *σ* distribution as well. The latter has been shown to obey Weibull’s statistics for such interfaces. However, whether it can also follow the most widely used normal (Gaussian) distribution is currently unclear. Moreover, a more complicated self-healing case, when the *σ* development at an initially amorphous interface is accompanied by its crystallization, has not been investigated yet in this respect. In order to address these two important issues, 10 pairs of amorphous poly(ethylene terephthalate) (PET) samples were kept in contact for various periods of time from 5 min to 15 h at a temperature (*T*) of 94 °C (preserving the amorphous state) or *T* = 150 °C (giving rise to cold crystallization), or both *T*s. Thereafter, the as-formed amorphous and semi-crystalline PET–PET auto-adhesive joints were shear fractured in tension at ambient temperature. For the first time, the statistical distributions of a number of the measured *σ* data sets were analyzed and discussed using both Weibull’s and the Gaussian model, including several normality tests.

## 1. Introduction

Physical self-bonding between the two contacting glassy polymer pieces at rather low temperatures (*T*), which are markedly below the glass transition temperature (*T*_g_) by some tens or even a hundred degrees of Kelvin, is an interesting phenomenon revealed by Boiko and Prud’homme at the end of the 1990s [1,2]. Its occurrence due to the chain interdiffusion from one contacting sample to another one with time (*t*), resulting in the creation of physical links of various natures (Van der Waals bonds, topological entanglements, etc.) at *T* < *T*_g_, is itself quite surprising since it was believed that this mode of molecular motion is frozen-in at such temperatures and is feasible at *T* > *T*_g_ only [3,4]. However, if *T*_g_ in a near-surface layer (*T*_g_^surface^) is decreased with respect to the bulk glass transition temperature (*T*_g_^bulk^), as it has been suggested in [5,6,7], then the long-range chain motions (e.g., reptation [8,9]) can be activated at *T*_g_^surface^ < *T* < *T*_g_^bulk^ and further persist at the interface as long as the interface *T*_g_ remains lower than *T*_g_^bulk^ [1,2,10].

The kinetics of the self-healing (or self-bonding) process at weak (due to its proceeding at rather low *T* < *T*_g_^bulk^) polymer–polymer interfaces have been analyzed extensively using the average lap shear strength value (*σ*_av_) [1,2,10]. It has been found that *σ*_av_ ~ *t*^1/4^ and log*σ*_av_ ~ 1/*T*, indicating that the self-healing process is a thermally activated process controlled by snake-like chain motions [8,9]. As further development of this field, the analysis of the statistical distribution of the *σ* values used to calculate the *σ*_av_ value and to estimate the data scatter, whether via the standard deviation (SD) of the mean or the so-called Weibull’s modulus (*m*) [11,12,13,14,15,16,17], can be performed. These data scatter parameters are helpful for a better understanding of the interface self-healing and fracture processes. Actually, it has been demonstrated that the distributions of the *σ* values that develop between two contacting pieces of similar or dissimilar amorphous and/or semi-crystalline polymers, such as amorphous polystyrene (PS), poly(methyl methacrylate) (PMMA), poly(2,6-dimethyl-1,4-phenylene oxide) (PPO), poly(ethylene terephthalate) (amPET), and semi-crystalline PET (crPET), obey Weibull’s statistics [18,19,20,21]. This implies that the linear plot in the specific coordinates “lnln[1/(1 − *P*_j_)] − ln*σ*” should be observed [11,12,13,14,15,16,17] (*P*_j_ is the cumulative probability of failure); the tangent of this curve *m* is a measure of the data scatter. This behavior is not surprising since it is characteristic of brittle and quasi-brittle materials [11,12,13,14,15,16,17] for which Weibull’s statistics were initially proposed [11] and to which weak (partially self-healed) polymer–polymer interfaces belong. The fracture process in such materials is controlled by the presence of surface defects such as surface cracks and their propagation across the sample cross-section upon sample fracture. The role of such defects in the fracture of partially (not fully) self-healed interfaces plays the interface plane, where the interpenetrated structure of the polymer bulk is not recovered.

Although the lap shear strength distributions of weak amorphous and semi-crystalline polymer–polymer interfaces have already been investigated in detail using Weibull’s statistics [18,19,20,21], currently, it is of interest to find out how the crystallization of amorphous polymer–polymer interfaces can impact Weibull’s statistical distribution behavior of *σ*. To put it differently, the question regarding the involvement of the phase transition in the strength development process in the course of self-healing arises. This more complicated case of interface self-healing has not been explored yet and, therefore, requires elucidation. By varying the crystallization conditions, one may expect a change in the *σ* statistical parameters (SD and *m*).

Furthermore, it is still not clear as whether the *σ* distributions of weak polymer–polymer interfaces, along with the Weibull’s distribution, also follow the most widely used normal (or Gaussian) distribution representing a bell curve [22], as it does, for instance, for the tensile strength distributions of high-strength polymer fibers (a type of statistical dualism) [17,23]. To put it differently, the manifestation of such dualism for the *σ* distributions of weak polymer–polymer interfaces is still an open question and clarification is important for further elucidation of the interface self-healing and fracture mechanisms.

So, the goal of this work is twofold: to find out, first, whether the lap shear strength variation of weak polymer–polymer interfaces obeys the normal distribution; and, second, how the crystallization of amorphous polymer–polymer interfaces induced in the course of self-healing impacts the type of *σ* statistical distribution, answering the question: which distribution type is most appropriate: Weibull’s, normal, or both of them?

To address these important issues, amorphous poly(ethylene terephthalate) (PET) was chosen since it can be produced both in the amorphous and semi-crystalline phase states. Three series of PET–PET self-bonding experiments were carried out. In the first series, 10 pairs of identical amorphous PET samples were kept in contact for a predetermined period of time (*t*) varying over a broad interval, from 5 min to 15 h, at a temperature (*T*) of 94 °C (> *T*_g_^bulk^). The choice of this elevated *T* was motivated by poor self-bonding ability of crystallizable PET at *T* < *T*_g_^bulk^ as compared to that of non-crystallizable PS and PPO [1,2] and the achievement of the highest *σ* values at the PET–PET interface while preserving the amorphous state [24]. In the second series, the amorphous PET–PET interfaces self-bonded at *T* = 94 °C as described above were further submitted to a higher *T* of 150 °C at which cold crystallization occurred, and kept at that temperature for the same period of *t* of 5 min. In the third series, the contacted amorphous PET samples were brought from room temperature to *T* = 150 °C and kept at that *T* from 5 min to 15 h. Thereafter, the as-self-bonded PET–PET interfaces were shear-fractured in tension at ambient temperature, and their lap shear strengths *σ* were measured. Finally, the distributions of the obtained *σ* data sets (totally, 11 sets) were analyzed regarding their conformity to Weibull’s and the Gaussian statistical models, including several normality tests [24,25].

## 2. Materials and Methods

### 2.1. Polymers and Samples

The polymer used in this study was commercial PET supplied by “Mogilevkhimvolokno”, Mogilev, Belarus. The thick bulk films of amorphous PET with a thickness of 100 μm were produced by compression molding of the PET pellets between Teflon films at *T* = 280 °C followed by rapid quenching in ice water. The as-produced amorphous PET films had a viscosity-average molecular weight of 15 kg/mol. The samples used had a width of 5 mm and a length of 30 mm.

### 2.2. Self-Bonding of PET–PET Interfaces

In order to form the PET–PET single lap shear autoadhesive joints (AJs) that are capable of bearing a mechanical load, 10 pairs of the amorphous PET samples were placed side by side in a flat aluminum assembly, brought into contact at an overlapped length of 5 mm, set in a *Carver* press and subjected to a small contact pressure of 0.4 MPa, and kept at a chosen self-bonding temperature *T* for a predetermined period of time *t*. Three different self-bonding protocols were used: (i) contact at *T* = 94 °C (above the amorphous PET *T*_g_^bulk^ = 81 °C [21]) for *t* = 5 min, 90 min, 4 h, and 15 h preserving the amorphous state followed by cooling down to room temperature (RT) by cold water circulating in the plates of the press; (ii) step (i) followed by heating up to *T* = 150 °C and exposition at this *T* for *t* = 5 min in all cases (this step gave rise to cold crystallization), and cooling down to RT; and (iii) contact at RT, submitting to heating up to *T* = 150 °C and exposition at this *T* for *t* from 5 min to 15 h followed by cooling down to RT.

### 2.3. Fracture Tests

The as-formed PET–PET AJs were shear-fractured in tension on an Instron tensile tester at RT at a crosshead speed of 5 mm/min. The distance between the tester clamps was 50 mm, with the joint located in the middle. The interface lap shear strength *σ* was calculated as the AJ fracture load *F* divided by the contact area *S* = 55 mm^2^. The procedures of the interface self-healing and fracture are schematically depictured in Figure 1.

### 2.4. Statistical Analysis

#### 2.4.1. Weibull’s Statistics

Weibull’s analysis was performed using Equation (1) [11,12,13,14,15,16,17]:lnln[1/(1 − *P*_j_)] = −*m*⋅ln*σ*_0_ + *m*⋅ln*σ*,(1)
where *P*_j_ = (*j* − 0.5)/*n* is the cumulative probability of failure, *n* is the joint number, *m* is the Weibull’s modulus (or the shape parameter), and *σ*_0_ is the scale parameter. Equation (1) can be simplified to:*y* = *a* + *bx*,(2)
where *y* = lnln[1/(1 − *P*_j_)], *b* is the *m*, *x* is the ln*σ*, and *a* = −*m*⋅ln*σ*_0_ is the curve intersect with the *y* axis. By estimating *m* as the tangent to the curve lnln[1/(1 − *P*_j_)] = f(ln*σ*) using the standard procedure of the linear regression analysis, one can calculate *σ*_0_ as *σ*_0_ = exp (−*a*/*m*).

#### 2.4.2. Normality Tests

To investigate the validity of the normal distribution, the *σ* data sets measured were analyzed, first, by constructing the normal probability (NP) or quantile-quantile (Q-Q) plots and, second, by computing them using several standard normality test procedures (Kolmogorov–Smirnov, Shapiro–Wilk, Lilliefors, Anderson–Darling, D’Agostino–K squared, and Chen–Shapiro tests) [24,25].

## 3. Results

In Figure 2, the values of the lap shear strength that developed at the symmetric amorphous and semi-crystalline PET–PET interfaces in the course of their partial self-healing at the three temperature regimes used (a total of 11 data sets) are shown. It is seen that the *σ* values for the PET–PET interfaces crystallized at *T* = 150 °C, either self-healed at *T* = 94 °C or not, overlap and are very close. Hence, whatever the durability of the self-healing step at *T* = 94 °C, it has a minor effect on the *σ* value that developed upon further exposition at *T* = 150 °C. This behavior can be explained by the fact that the samples used in these two different self-healing series, including the exposition at *T* = 150 °C, had the same crystallinity index *C* = 0.33, as estimated by a DSC method, not depending on *t* and the self-healing prehistory (*T* = 94 °C + 150 °C or *T* = 150 °C) [20]. The *σ* values that developed at the PET–PET interfaces in the amorphous state at *T* = 94 °C are lower with respect to those that developed at the interfaces in the semi-crystalline state. However, this difference is rather small (about 25% at *t* = 15 h). This means that the major contribution to the developed lap shear strength is provided by the physical links (van der Waals bonds) originated between the molecular groups of the chain segments diffused across the amorphous interface and those of the counter surface polymer matrix.

The 11 data sets presented in Figure 2 were investigated regarding their conformity to both the normal and Weibull’s distributions. For this purpose, the following four plot types were constructed (see Figure 3, Figure 4, Figure 5, Figure 6, Figure 7, Figure 8, Figure 9, Figure 10, Figure 11, Figure 12 and Figure 13): (a) Weibull’s plots lnln[1/(1 − *P*_j_)] vs. ln*σ*, (b) normal probability plots (normal percentiles (NP) vs. *σ*), (c) probability density function (PDF) vs. *σ*, and (d) quantile–quantile (Q–Q) plots (expected normal value vs. measured value; only in Figure 3d). The results of this analysis for self-bonding at (i) *T* = 94 °C, (ii) *T* = 94 °C followed by *T* = 150 °C, and (iii) *T* = 150 °C are presented in Figure 3, Figure 4, Figure 5 and Figure 6, Figure 7, Figure 8, Figure 9 and Figure 10, and Figure 11, Figure 12 and Figure 13, respectively.

As follows from Figure 3a, Figure 4a, Figure 5a, Figure 6a, Figure 7a, Figure 8a, Figure 9a, Figure 10a, Figure 11a, Figure 12a, Figure 13a and Table 1, all the Weibull’s plots constructed are linear and characterized by rather high values of the root mean square deviation *R*^2^ > 0.95. Moreover, as follows from Table 1, the values of the ratio of the scale parameter *σ*_0_ estimated from the Weibull’s plots to the average strength *σ*_av_, *σ*_0_/*σ*_av_, are close to 1 for all 11 self-bonding conditions used. Therefore, the Weibull’s fitting results obtained are correct in all the cases considered. As far as the values of the Weibull’s modulus *m* are concerned (see Figure 14a), they vary over a rather broad interval, from *m* = 3.07 (*T* = 150 °C, *t* = 5 min) to *m* = 16.27 (*T* = 94 °C, *t* = 5 min). Since the *m* value is a measure of the data scatter, the smallest and the largest data scatters are characteristic of the shortest time investigated *t* = 5 min at the amorphous non-crystallized and semi-crystalline (*C =* 0.33) PET–PET interfaces, respectively. However, the *m* values for the three different self-bonding regimes considered converge with an increase in *t*. Moreover, at the longest investigated *t* = 15 h, the largest value *m* = 13 is calculated after crystallization at *T* = 150 °C. To put it differently, a negative role of crystallization from the point of view of the data scatter observed at short times vanishes at long times. More specifically, with an increase in *t* from 5 min to 15 h, the following trends are observed for the three self-bonding regimes used.

After self-bonding at *T* = 94 °C, the *m* value decreases. When the self-bonding step of the exposition of the PET–PET interface at *T* = 150 °C for 5 min is added to the step of its exposition at *T* = 94 °C, the *m* value, itself weakly dependent on *t*, is intermediate between those estimated after the self-bonding at *T* = 94 °C and at *T* = 150 °C. This seems to be reasonable for this intermediate self-bonding regime fully including the first one and partially the third one. These observations may be rationalized as follows.

During the early self-healing stages of the amorphous PET–PET interface at *T* = 94 °C (*t* = 5 min), the chain end interdiffusion process begins, proceeding rather uniformly at its beginning. However, when it progresses with *t* with the corresponding increase in *σ*_av_ (from 0.21 to 0.30 MPa), the interface interpenetrated structure becomes more entangled and less relaxed and equilibrated, which results in the observed increase in the data scatter (a decrease in *m*). After the self-bonding at *T* = 150 °C (including the heating step from room temperature to *T* = 150 °C for ~2 min) accompanied with crystallization (formation of ordered crystallites), the *m* value increases substantially with an increase in *t* from 5 min to 15 h from *m* = 3.07 to *m* = 13.09, with the corresponding marked decrease in the data scatter, indicating that the negative role of crystallization diminishes with *t*. It is interesting to note that this effect is observed for the samples with the same average lap shear strength of *σ*_av_ = 0.36–0.37 MPa and the same crystallinity of 0.33. These observations suggest that the chain interdiffusion process does not progress with *t* at *T* = 150 °C while, simultaneously, the interface structure becomes more uniform and equilibrated in the course of this “annealing” with an increase in *t*. To put it differently, the arrest of self-healing by the chain interdiffusion is accompanied by a more uniform crystallization (more regular chain folding).

Comparing the *m* values presented in Figure 14a varying over a rather broad interval (*m* = 2–17) with those reported earlier [18,19,20,21] for the polymer–polymer interfaces with at least one PET sample of the two contacted samples, the *m* values for the amPET–amPET (*m* = 5–17) [19], amPET–crPET (*m* = 2–14) [20], crPET–crPET (*m* = 4–10) [20], and amPS–amPET (*m* = 3–11) interfaces are identified [21]. It is seen that the *m* value intervals for these four PET-containing interfaces also vary over rather broad intervals, which overlap with that estimated in the present work. So, it may be concluded that crystallization of the amPET–amPET interface does not impact on the data scatter. Moreover, it is interesting to note that close *m* values were observed for the two types of crPET–crPET interfaces: when the contacting samples were crystallized prior to contact (*m* = 4–10) [20] or in the course of contact (present work, *m* = 3–13). Therefore, the data scatter for the partially self-healed crPET–crPET interfaces seems to be weakly dependent on the self-healing stage when crystallites were formed. However, it should be noted that the highest value *σ* = 0.1 MPa achieved at the interface between the two PET samples crystallized prior to contact [20] is substantially, by a factor of 5, lower compared to the highest value *σ* = 0.5 MPa achieved after the self-healing in the amorphous state followed by cold crystallization (see Figure 2). This substantial difference in the *σ* values attainable at these two crPET–crPET interfaces indicates that the healing step in the amorphous state prior to crystallization is an effective way to improve the adhesion strength at the crPET–crPET interfaces.

It should be noted that the comparable intervals of the *m* variation were reported for the lap shear strength of the weak (partially self-healed) PS–PS (*m* = 1–4), PMMA–PMMA (*m* = 1–5), and PS–PMMA (*m* = 6–11) interfaces [18], and for completely different types of materials such as high-performance polymer fibers characterized by a very high tensile strength *σ* = 0.5–5 GPa (*m* = 7–10) [17,23]. This behavior suggests similar fracture mechanisms of these two completely different polymer systems consisting in their brittleness.

In order to investigate the validity of the normal distribution for the lap shear strength variation at the amorphous and semi-crystalline PET–PET interfaces, analysis of the *σ* data sets presented in Figure 2 using both the NP and Q–Q plots was carried out. Along with receiving the response concerning their linearity, this analysis also provides the values of *σ*_av_ and SD for each set of measurements (see Table 2), which is a simple and convenient way to estimate those parameters just by putting a set of the measured *σ* values into a computing system to draw the NP or Q–Q plots. As it was expected a priori, the linear fitting results for these two plots coincide (compare Figure 3b,d), with both of them having a linear shape, and the identical values of *σ*_av_ and SD were estimated for the two plot types. For this reason, only one of the two normal plots, the NP plot, was used for the analysis in the present work.

When analyzing the SD variation with *t* (see Figure 14b), one may notice a reversed (mirror-like) trend with respect to that observed for *m* (see Figure 14a) since an increase in *m* means a decrease in the data scatter while it means its increase for SD. Hence, each of these two statistical parameters can be used for characterization of the data scatter. However, for the purpose of comparison of the data scatter for the materials that drastically differ in their mechanical properties, it seems to be preferable to operate with the Weibull’s modulus than with SD since the former and the latter are expressed in arbitrary and absolute units, respectively. Therefore, the use of *m* is more convenient because it allows one to compare the distribution behaviors of materials differing drastically in strength (e.g., *m* ≈ 5–10 for both weak polymer–polymer interfaces with *σ* = 0.01–0.5 MPa [18,19,20,21] and ultra-high-strength fibers with *σ* = 2–6 GPa [17,23]). By contrast, such a comparison is not straightforward when using SD (±0.01 and ±100 MPa, respectively) since roughly the same difference in the *σ* values (by four orders of magnitude) is retained for the SD values as well. To put it differently, the Weibull’s modulus seems to be a more universal statistical parameter characterizing the data scatter, which can be recommended for its more extensive use for characterization of the materials of the different levels of strength and phenomena of various natures.

In order to investigate a possible correlation between SD and *m*, the data of Figure 14a,b are considered in Figure 15 as SD vs. *m*. As shown, the resulting united graph can be fitted with an asymptotic master curve described by Equation (3), which can be used for conversion of SD to *m*, and vice versa:SD = −0.0264 + 0.187·0.908 ^m^.(3)

To investigate the normality of the data sets shown in Figure 2, the NP plots for the 11 self-healing conditions used were constructed; they are presented in Figure 3b, Figure 4b, Figure 5b, Figure 6b, Figure 7b, Figure 8b, Figure 9b, Figure 10b, Figure 11b, Figure 12b, Figure 13b. As shown, all the NP plots have a linear shape, indicating that the normal distribution is expected in all the cases considered. Nevertheless, in order to further confirm the validity of this conclusion, in addition to the analysis using the NP plots, the *σ* distributions under investigation were also analyzed using several standard normality tests such as the Kolmogorov–Smirnov, Shapiro–Wilk, Lilliefors, Anderson–Darling, D’Agostino–K squared, and Chen–Shapiro tests [24,25]. The statistical results of those tests are collected in Table 3 and Table 4. The results of the analysis carried out have led to the conclusion that the normality “cannot be rejected” for all the data sets reported in Figure 3b, Figure 4b, Figure 5b, Figure 6b, Figure 7b, Figure 8b, Figure 9b, Figure 10b, Figure 11b, Figure 12b, Figure 13b using each of the normality tests indicated above. Therefore, the bell curves characteristic of the Gaussian distribution are expected when replotting these experimental data sets as the probability density function (PDF) vs. *σ*. It follows from Figure 3c, Figure 4c, Figure 5c, Figure 6c, Figure 7c, Figure 8c, Figure 9c, Figure 10c, Figure 11c, Figure 12c, Figure 13c that the majority of the histograms PDF vs. *σ* constructed (three quarters) can be fitted with the bell-shaped curves while some of the curves (e.g., the fitting curves in Figure 8c, Figure 9c, Figure 13c, i.e., a quarter of the curves analyzed) do not demonstrate the bell shape.

The results on the conformity of the lap shear strength distributions of the PET–PET interfaces investigated to the Weibull’s and Gaussian models estimated with the help of the statistical analysis carried out above are compared in Table 5. They demonstrate that the Weibull’s model works correctly in all the cases analyzed (*R*^2^ > 0.95) while the Gaussian model does for roughly a half of the curves analyzed. Therefore, the result “cannot reject normality” obtained in the normality tests performed does not necessarily mean that the data can be correctly described with the bell-shaped curve. Nevertheless, in general, the majority of the data sets can be considered as formally satisfying the normal distribution. Therefore, it may be concluded that Weibull’s model seems to be more appropriate to correctly describe the statistical distributions of *σ* that developed both at the amorphous and semi-crystalline PET–PET interfaces. Nevertheless, the Gaussian distribution can also be recommended for the data scatter analysis of weak polymer–polymer interfaces as a complimentary statistical estimate of *σ*.

Despite the fact that the Weibull’s model works better than the Gaussian one, the latter, including the normality tests, should also be involved in the statistical analysis of the lap shear strength distributions, at the amorphous and semi-crystalline PET–PET interfaces in particular, and of the tensile strength of a tremendous number of various polymer and composite materials. Actually, the validity of the Weibull’s model suggests that the interface fracture process is initiated by the interface defects, the role of which is played by the interface plane. The interfaces investigated were weak (*σ* < 0.6 MPa) since they were only partially self-healed (for comparison, a fully self-healed amorphous PS–PS interface is characterized by *σ* = 7.2 MPa [9], which is an order of magnitude larger). In this case, the interface itself represents the major crack propagation after application of the mechanical load. Moreover, the validity or non-validity of the normal distribution can provide additional information on the interface fracture mechanism since, in the first case, it implies that the fracture process is influenced by the sum of many independent and equally weighted factors, e.g., such as the uniformity of the interpenetration depth upon the self-healing interface, and chain pullout and bond rupture upon the interface fracture.

## 4. Conclusions

For the first time, the statistical distributions of the development of lap shear strength *σ* during the early self-bonding stages of initially amorphous or partially crystallized thereafter PET–PET interfaces were investigated regarding their conformities to the Weibull’s and normal distributions. Weibull’s model was found to be correct for all 11 analyzed *σ* distribution curves both for the amorphous and semi-crystalline PET–PET interfaces, in strong agreement with the *σ* statistical distribution behavior for the weak interfaces of other similar and dissimilar amorphous non-crystallizable polymers (PS, PMMA, PPO, PS-PPO blends) reported earlier [18,19,20,21], including the close values of *m*. This fact confirms the validity of Weibull’s statistics to correctly describe the adhesion strength distribution at weak polymer–polymer interfaces, which can be recommended for further use in such polymer systems. Moreover, it indicates that the interface is quasi-brittle, and its fracture process is controlled by the propagation of the major crack, the role of which is played by the interface itself.

The impact of crystallization induced in the course of the self-bonding process of the initially amorphous PET samples on the Weibull’s parameters of the *σ* distribution was investigated. It was shown that the variation of the Weibull’s modulus *m* depended on the time–temperature self-bonding conditions and the phase state (amorphous or semi-crystalline) of the PET–PET interfaces. For the three self-bonding temperature regimes used, the following impact of an increase in the contact time *t* from 5 min to 15 h on the *m* value was found: the value of *m* (i) decreased after the self-bonding at *T* = 94 °C in the amorphous state, (ii) remained nearly constant after the self-bonding at *T* = 94 °C in the amorphous state followed by cold crystallization at *T* = 150 °C, and (iii) increased markedly after the self-bonding at *T* = 150 °C simultaneously accompanied by cold crystallization. These three findings give further important details on the self-bonding process statistics, indicating that the interpenetrated interface structure developed via the chain interdiffusion becomes more entangled and less equilibrated when the interface self-healing process occurs in the amorphous state. In a semi-crystalline state, the interdiffusion is arrested due to the formation of crystallites, wherein the motions of the chain portions are blocked. Simultaneously, the interface crystalline structure can become more regular and uniform with the contact time.

Each of the normality tests performed (totally, 94) provided the result “cannot reject normality”, indicating that all the data sets follow the normal distribution. These results were confirmed by the observation of the clearly defined bell curves, which are characteristic of the Gaussian (normal) distribution on the majority (three quarters) of the analyzed “PDF−*σ*” histograms. Nevertheless, the formal validity of normality confirmed in the normality tests does not necessarily mean that all the lap shear strength distributions should represent the bell curves PDF(*σ*), as it does for the remaining part (a quarter) of these histograms. To put it differently, the final conclusion concerning the correspondence or non-correspondence of the *σ* distribution to the Gaussian distribution can be made only after the analysis of the PDF−*σ* curves. The best conformity of the lap shear strength distributions to the Gaussian model was found for non-crystallized interfaces, indicating that the interface self-bonding process in this phase state is controlled, on the one hand, by the sum of many independent and equally weighted factors, and, on the other hand, by the crack propagation across the quasi-brittle interface, as it follows from the Weibull-like behavior. These are the signs of statistical dualism observed for high-performance polymer materials earlier [17,23].

Since both the Gaussian and Weibull’s models are complimentary, the combined Gaussian/Weibull’s analysis is recommended for characterization of the statistical distribution of the self-bonding strength developed at weak polymer–polymer interfaces. The Weibull’s modulus can be considered as a preferable measure for characterizing the data scatter compared to the standard deviation since the former is a dimensionless statistical parameter that makes it possible to carry out a correct comparison of the data scatter for materials drastically differing in strength (by some order of magnitude). The Weibull’s modulus seems to be useful for investigating the uniformity of the adhesion strength evolution and the stability and completeness of the self-bonding process. 

Finally, for a better understanding of the physical processes of various natures (e.g., fracture, adhesion, interface self-healing etc.) in polymers and other materials it is recommended to operate not only with the average value of a property under analysis but with its statistical distribution as well. This approach can give additional useful information and new insight into the molecular mechanisms controlling these processes which can be masked when considering only the average values characterizing them. 

## Figures and Tables

**Figure 1 polymers-14-04519-f001:**
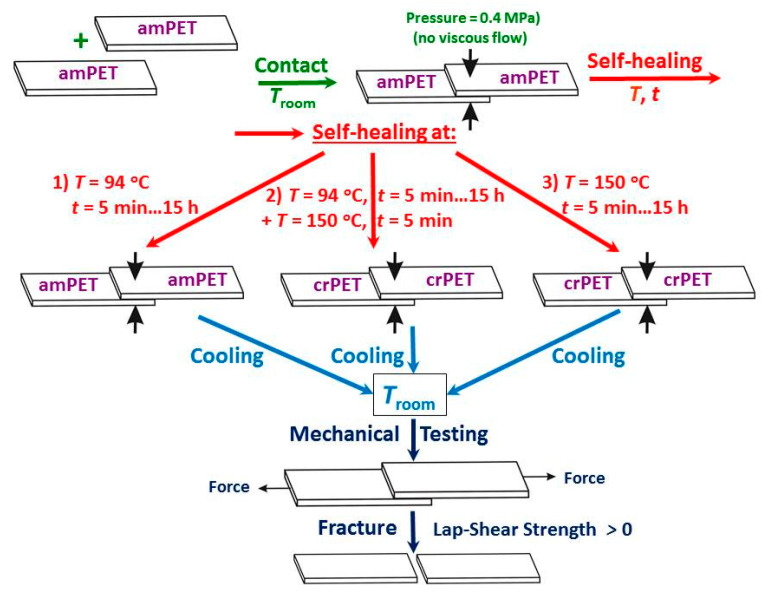
Schematic representation of the experimental procedures used in this work.

**Figure 2 polymers-14-04519-f002:**
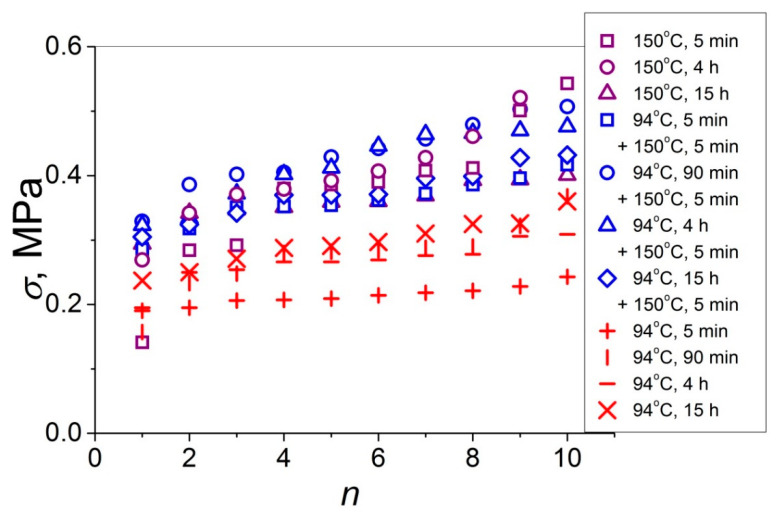
Lap-shear strength *σ*, shown as a function of the joint number in ascending order, that developed at several symmetric PET–PET interfaces: (crosses and dashes) amorphous interface self-healed at *T* = 94 °C for *t* = 5 min to 15 h, (open blue symbols) amorphous interface self-healed at *T* = 94 °C for *t* = 5 min to 15 h and submitted thereafter to self-healing and cold crystallization at *T* = 150 °C for *t* = 5 min, and (open purple symbols) amorphous interface submitted to self-healing and cold crystallization at *T* = 150 °C for *t* = 5 min to 15 h.

**Figure 3 polymers-14-04519-f003:**
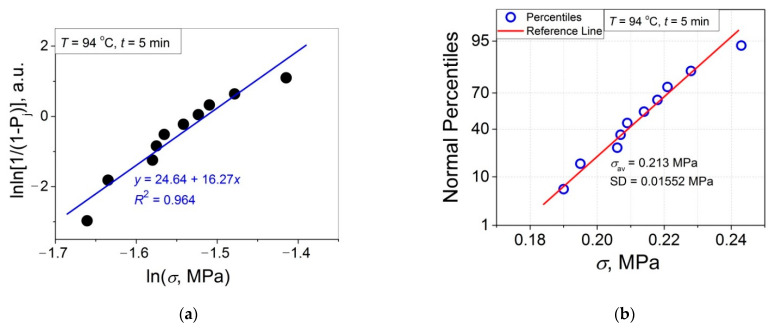
(**a**) Linear regression fit of the lap shear strength *σ* in Weibull’s coordinates; (**b**) normal probability plot; (**c**) probability density function (PDF) vs. *σ* fitted with Gaussian function; (**d**) normal quantile–quantile (Q–Q) plot for an amorphous PET–PET interface self-bonded at *T* = 94 °C for *t* = 5 min.

**Figure 4 polymers-14-04519-f004:**
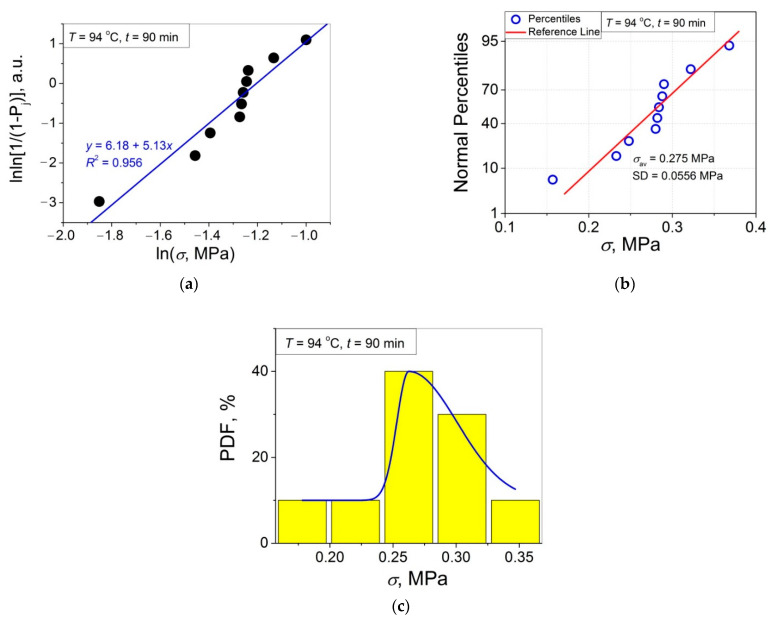
(**a**) Linear regression fit of the lap shear strength *σ* in Weibull’s coordinates; (**b**) normal probability plot; (**c**) probability density function (PDF) vs. *σ* fitted with a bi-Gaussian function for an amorphous PET–PET interface self-bonded at *T* = 94 °C for *t* = 90 min.

**Figure 5 polymers-14-04519-f005:**
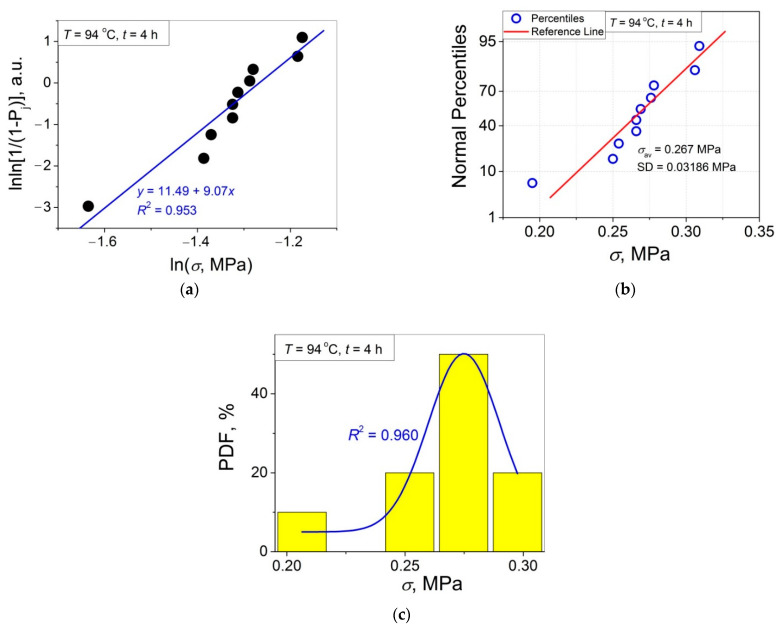
(**a**) Linear regression fit of the lap shear strength *σ* in Weibull’s coordinates; (**b**) normal probability plot; (**c**) probability density function (PDF) vs. *σ* fitted with a Gaussian function for an amorphous PET–PET interface self-bonded at *T* = 94 °C for *t* = 4 h.

**Figure 6 polymers-14-04519-f006:**
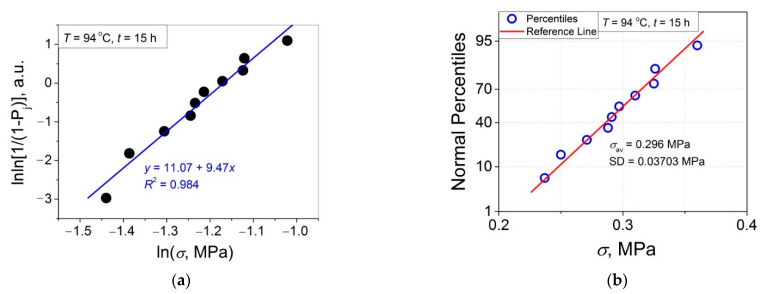
(**a**) Linear regression fit of the lap shear strength *σ* in Weibull’s coordinates; (**b**) normal probability plot; (**c**) probability density function (PDF) vs. *σ* fitted with a Gaussian (solid line) and bi-Gaussian functions for an amorphous PET–PET interface self-bonded at *T* = 94 °C for *t* = 15 h.

**Figure 7 polymers-14-04519-f007:**
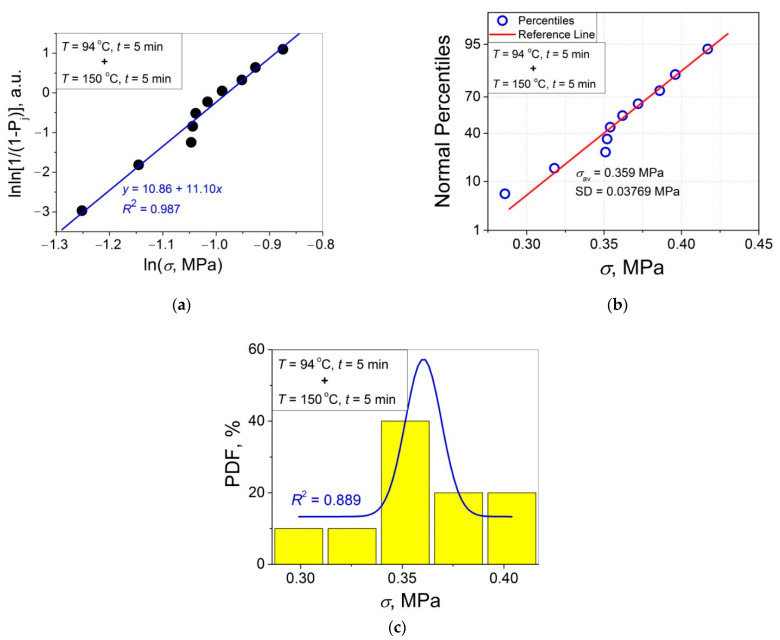
(**a**) Linear regression fit of the lap shear strength *σ* in Weibull’s coordinates; (**b**) normal probability plot; (**c**) probability density function (PDF) vs. *σ* fitted with a Gaussian function for an amorphous PET–PET interface self-bonded at *T* = 94 °C for *t* = 5 min and at *T* = 150 °C for *t* = 5 min.

**Figure 8 polymers-14-04519-f008:**
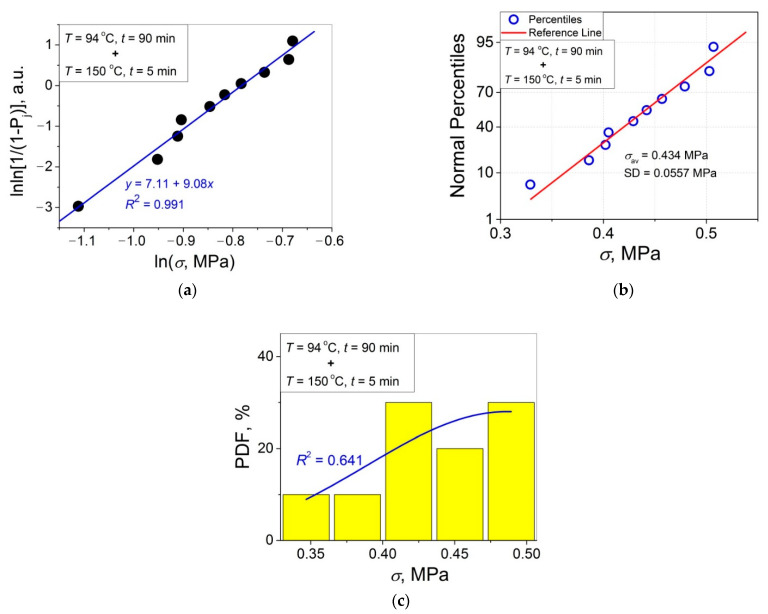
(**a**) Linear regression fit of the lap shear strength *σ* in Weibull’s coordinates; (**b**) normal probability plot; (**c**) probability density function (PDF) vs. *σ* fitted with a Gaussian function for an amorphous PET–PET interface self-bonded at *T* = 94 °C for *t* = 90 min and at *T* = 150 °C for *t* = 5 min.

**Figure 9 polymers-14-04519-f009:**
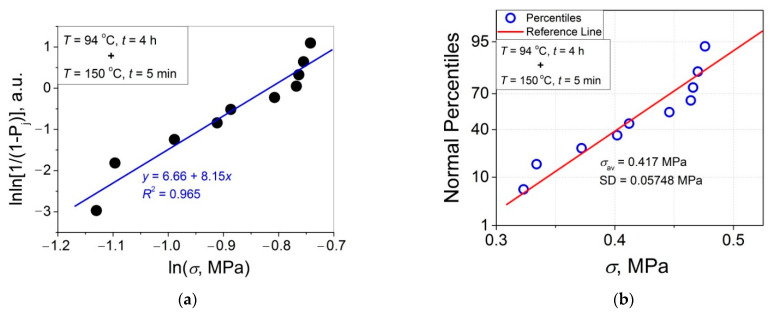
(**a**) Linear regression fit of the lap shear strength *σ* in Weibull’s coordinates; (**b**) normal probability plot; (**c**) probability density function (PDF) vs. *σ* fitted with a Gaussian function for an amorphous PET–PET interface self-bonded at *T* = 94 °C for *t* = 4 h and at *T* = 150 °C for *t* = 5 min.

**Figure 10 polymers-14-04519-f010:**
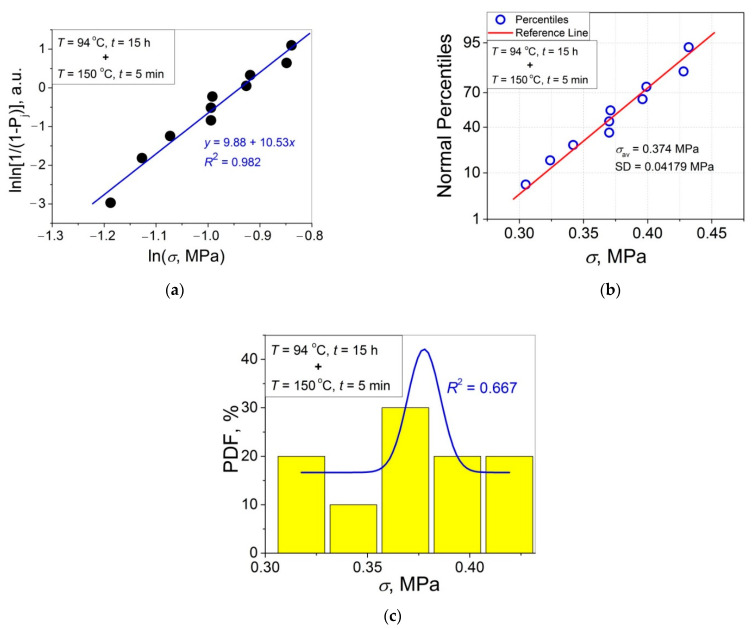
(**a**) Linear regression fit of the lap shear strength *σ* in Weibull’s coordinates; (**b**) normal probability plot; (**c**) probability density function (PDF) vs. *σ* fitted with a Gaussian function for an amorphous PET–PET interface self-bonded at *T* = 94 °C for *t* = 15 h and at *T* = 150 °C for *t* = 5 min.

**Figure 11 polymers-14-04519-f011:**
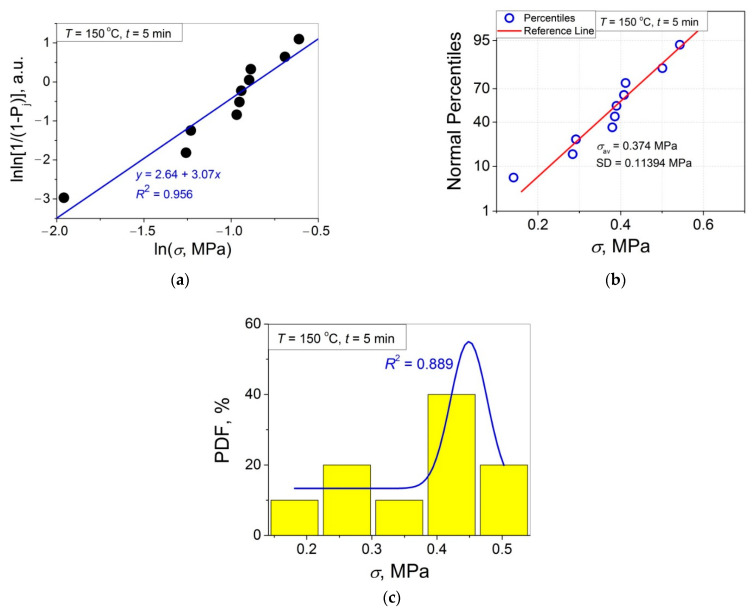
(**a**) Linear regression fit of the lap shear strength *σ* in Weibull’s coordinates; (**b**) normal probability plot; (**c**) probability density function (PDF) vs. *σ* fitted with a Gaussian function for an amorphous PET–PET interface self-bonded at *T* = 150 °C for *t* = 5 min.

**Figure 12 polymers-14-04519-f012:**
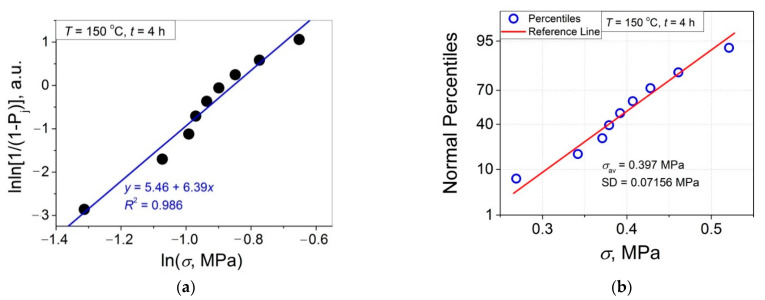
(**a**) Linear regression fit of the lap shear strength *σ* in Weibull’s coordinates; (**b**) normal probability plot; (**c**) probability density function (PDF) vs. *σ* fitted with a Gaussian function for an amorphous PET–PET interface self-bonded at *T* = 150 °C for *t* = 4 h.

**Figure 13 polymers-14-04519-f013:**
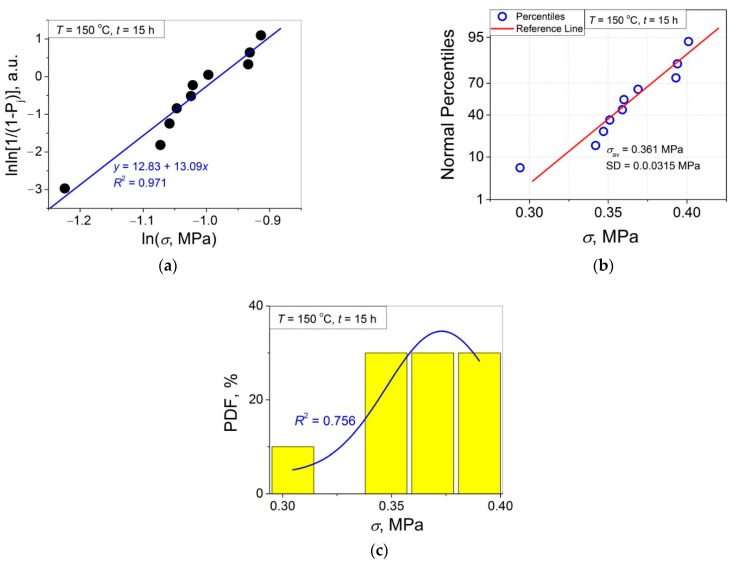
(**a**) Linear regression fit of the lap shear strength *σ* in Weibull’s coordinates; (**b**) normal probability plot; (**c**) probability density function (PDF) vs. *σ* fitted with a Gaussian function for an amorphous PET–PET interface self-bonded at *T* = 150 °C for *t* = 15 h.

**Figure 14 polymers-14-04519-f014:**
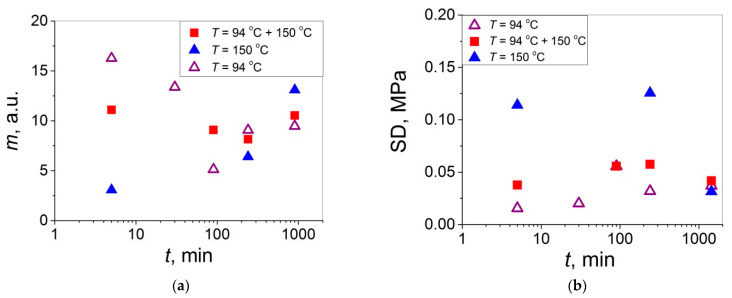
(**a**) Weibull’s modulus *m* and (**b**) standard deviation SD as a function of the healing time *t* for initially amorphous symmetric PET–PET interfaces self-healed: (open triangles) at *T* = 94 °C, (solid squares) self-healed at *T* = 94 °C and thereafter at *T* = 150 °C for 5 min, and (solid triangles) submitted to self-healing at *T* = 150 °C.

**Figure 15 polymers-14-04519-f015:**
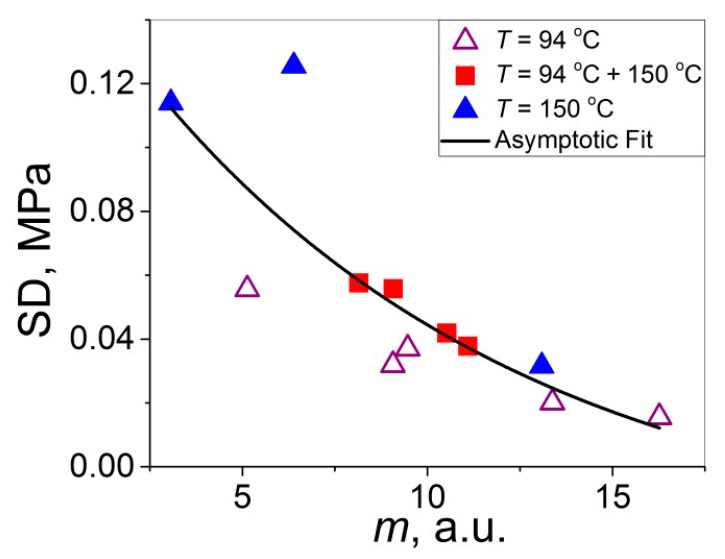
Data of Figure 14a,b plotted as SD vs. *m* fitted with an asymptotic curve SD = −0.0264 + 0.187·0.908 ^m^ (*R*^2^ = 0.603).

**Table 1 polymers-14-04519-t001:** Results of the Weibull’s analysis of the lap shear adhesion strength distribution for amPET–amPET interfaces self-healed (i) at *T* = 94 °C for various intervals of healing time, (ii) at *T* = 94 °C for various intervals of healing time followed by an additional self-healing step at *T* = 150 °C for 5 min, and (iii) at *T* = 150 °C for various *t*.

*T*, °C (Phase State)	*t*, min	*y* = *a* + *bx*	*R* ^2^	*m*	*σ*_0_, MPa	*σ*_av_, MPa	*σ*_0_/*σ*_av_
(i) 94 (am.)	5	*y* = 24.64 + 16.27*x*	0.964	16.27	0.22	0.21	1.03
	90	*y* = 6.18 + 5.13*x*	0.956	5.13	0.30	0.28	1.09
	240	*y* = 11.49 + 9.07*x*	0.964	9.07	0.28	0.27	1.06
	900	*y* = 11.07 + 9.47*x*	0.984	9.47	0.31	0.30	1.05
(ii) 94 → 150	5	*y* = 10.86 + 11.10*x*	0.987	11.10	0.38	0.36	1.06
(am. → cr.)	90	*y* = 7.11 + 9.08*x*	0.991	9.08	0.46	0.43	1.07
	240	*y* = 6.66 + 8.15*x*	0.966	8.15	0.44	0.42	1.05
	900	*y* = 9.88 + 10.53*x*	0.982	10.53	0.39	0.37	1.05
(iii) 150	5	*y* = 2.64 + 3.07*x*	0.956				
				3.07	0.42	0.37	1.14
(am. → cr.)	90	*y* = 5.46 + 6.39*x*	0.986	6.39	0.43	0.40	1.08
	240	*y* = 12.83 + 13.09*x*	0.965	13.09	0.38	0.36	1.06

**Table 2 polymers-14-04519-t002:** NP or Q–Q plots of the statistical results (*σ*_av_ and SD) estimated for PET–PET interfaces.

Phase State	*T*, °C	*t*, min	*σ*_av_, MPa	SD, MPa
	94	5	0.213	0.01552
am		30	0.231	0.02009
		90	0.275	0.0556
		240	0.267	0.03186
		1440	0.296	0.03703
am → cr	94 + 150	5	0.359	0.03769
		90	0.434	0.0557
		240	0.417	0.05748
		1440	0.374	0.04179
		5	0.374	0.11394
am → cr	150	240	0.397	0.07156
		1440	0.361	0.0315

**Table 3 polymers-14-04519-t003:** Statistical parameters of the lap shear strength distribution for PET–PET interfaces estimated in several normality tests.

*T*, °C	*t*, min	Test Type	Statistic	*p*-Value	Decision at Level 5%
94	5	Kolmogorov–Smirnov	0.12371	1	+
	30		0.15207	1	+
	90		0.2344	0.57673	+
	240		0.1979	0.80775	+
	1440		0.11975	1	+
94 + 150	5	Kolmogorov–Smirnov	0.21182	0.7157	+
	90		0.09805	1	+
	240		0.19609	0.8201	+
	1440		0.16472	1	+
150	5	Kolmogorov–Smirnov	0.22205	0.65102	+
	240		0.23293	0.58531	+
	1440		0.17317	0.98786	+
94	5	Shapiro–Wilk	0.97685	0.94612	+
	30		0.94385	0.59665	+
	90		0.92466	0.39744	+
	240		0.89299	0.18322	+
	1440		0.98188	0.97441	+
94 + 150	5	Shapiro–Wilk	0.96525	0.84368	+
	90		0.96334	0.82321	+
	240		0.87861	0.12577	+
	1440		0.95535	0.7318	+
150	5	Shapiro–Wilk	0.93964	0.54907	+
	240		0.86346	0.08382	+
	1440		0.91948	0.35266	+
94	5	Lilliefors	0.12371	0.2	+
	30		0.15207	0.2	+
	90		0.2344	0.57673	+
	240		0.1979	0.2	+
	1440		0.19609	0.2	+
94 + 150	5	Lilliefors	0.21182	0.2	+
	90		0.09805	0.2	+
	240		0.07514	0.96313	+
	1440		0.16472	0.2	+
150	5	Lilliefors	0.22205	0.16085	+
	240		0.23293	0.12518	+
	1440		0.17317	0.2	+
94	5	Anderson–Darling	0.16698	0.91062	+
	30		0.24647	0.67612	+
	90		0.47624	0.18345	+
	240		0.49714	0.16077	+
	1440		0.14785	0.94581	+
94 + 150	5	Anderson–Darling	0.2428	0.68968	+
	90		0.17448	0.89695	+
	240		0.49025	0.16797	+
	1440		0.22692	0.74779	+
150	5	Anderson–Darling	0.36767	0.35571	+
	240		0.61003	0.08044	+
	1440		0.36385	0.36383	+
		D’Agostino–K squared:			
94		Omnibus	0.64785	0.7233	+
	5	Skewness	0.6545	0.51279	+
		Kurtosis	0.46849	0.63944	+
		Omnibus	1.12627	0.56942	+
	30	Skewness	−0.01862	0.98515	+
		Kurtosis	−1.0611	0.28865	+
		Omnibus	3.06324	0.21619	+
	90	Skewness	−1.01026	0.31237	+
		Kurtosis	1.4292	0.15295	+
		Omnibus	5.07418	0.0791	+
	240	Skewness	−1.52683	0.1268	+
		Kurtosis	1.65619	0.09768	+
		Omnibus	0.00427	0.99787	+
	1440	Skewness	0.06528	0.94795	+
		Kurtosis	0.00228	0.99818	+
94 + 150		Omnibus	1.01609	0.60167	+
	5	Skewness	−0.80253	0.42225	+
		Kurtosis	0.60995	0.54189	+
		Omnibus	0.36197	0.83445	+
	90	Skewness	−0.59871	0.54937	+
		Kurtosis	0.05935	0.95267	+
		Omnibus	1.73848	0.41927	+
	240	Skewness	−0.93093	0.35189	+
		Kurtosis	−0.93373	0.34045	+
		Omnibus	0.29125	0.86448	+
	1440	Skewness	−0.26825	0.78851	+
		Kurtosis	−0.46829	0.63958	+
150		Omnibus	1.72413	0.42229	+
	5	Skewness	−0.96005	0.33703	+
		Kurtosis	0.89579	0.37037	+
		Omnibus	0.59618	0.74223	+
	240	Skewness	−0.02926	0.97666	+
		Kurtosis	0.77157	0.44037	+
		Omnibus	2.43812	0.29551	+
	1440	Skewness	−1.17579	0.23968	+
		Kurtosis	1.02744	0.30421	+

“+” in column 6 means “cannot reject normality”.

**Table 4 polymers-14-04519-t004:** Statistical parameters of the lap shear strength distribution for PET–PET interfaces estimated from the Chen–Shapiro normality test.

*T*, °C	*t*, min	Statistic	10% Critical Value	5% Critical Value	Decision at Level 5%
94	5	−0.16927	0.01981	0.06668	+
	90	−0.07903	0.01981	0.06668	+
	240	−0.02348	0.01981	0.06668	+
	1440	−0.17618	0.01981	0.06668	+
94 + 150	5	−0.14999	0.01981	0.06668	+
	90	−0.14393	0.01981	0.06668	+
	240	0.00485	0.01981	0.06668	+
	1440	−0.13387	0.01981	0.06668	+
150	5	−0.10557	0.01981	0.06668	+
	240	0.03057	0.01981	0.06668	+
	1440	−0.06635	0.01981	0.06668	+

“+” in column 6 means “cannot reject normality”.

**Table 5 polymers-14-04519-t005:** Conformity of the lap shear strength distributions of PET–PET interfaces to Weibull’s and Gaussian models.

Self-Bonding Temperature *T*, °C	Phase State	Self-Bonding Time *t*, min	Gaussian Fitting (*R*^2^)	Weibull’s Fitting (*R*^2^)
94	amorphous	5	0.931	0.964
		90	± *	0.956
		240	0.96	0.964
		1440	0.889	0.984
94 + 150	amorphous → semi-crystalline	5	0.889	0.987
		90	±(0.641)	0.991
		240	−0.804	0.966
		1440	±(0.667)	0.982
150	amorphous → semi-crystalline	5	0.889	0.956
		240	1	0.986
		1440	±(0.756) **	0.965

* Can be fitted with a bi-Gaussian function; ** For Gaussian fitting, “±” means that the bell curve is observed qualitatively but the fitting reliability is poor.

## Data Availability

Not applicable.

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
