# Peer review of "Impact of Crystallization on the Development of Statistical Self-Bonding Strength at Initially Amorphous Polymer–Polymer Interfaces"

_polymers, 2022, doi:10.3390/polym14214519_

Round 1

Reviewer 1 Report

This is a sound work, and the manuscript is well-written.  In this work, the author investigated self-bonding strength developed at initially amorphous polymer-polymer interfaces using amorphous PET as a benchmark. More importantly, the author applied experimental data into Weibull's and Gaussian statistical models and concluded these two models are complimentary. This work should provide very useful experimental and theoretical insights to the understanding of polymer self-bonding process. I would recommend direct publication. 

Author Response

Responses to Reviewer 1

Many thanks for your positive evaluation of this work.

Reviewer 2 Report

Impact of Crystallization on the Statistical Self-Bonding Strength Developed at Initially Amorphous Polymer-Polymer Interfaces 

Boiko explores mechanisms of adhesion strength development during the initial stages of self-healing of polymer-polymer interfaces. Further, they use statistical analysis (particularly) Weibull’s and Gaussian models to fit several statistical distributions from various datasets to understand which type of model describes the data the best. Overall, the study is interesting, and the observations are stimulating. The author has published several papers in this field and is an expert in statistical analysis of the data. However, there are some sample processing concerns that need to be addressed before this can be published in Polymers. Since the idea is to understand the effect of crystallization on the development of statistical self-bonding strength, it is crucial to ensure a deeper understanding of the crystallization process. Below are some of the comments in no particular order of importance: 

(1)  The processing that has been used to produce amorphous pellets is not well controlled. They say the samples were rapidly quenched from 280 C to ice bath temperature. Such rapid quenching can result in void formation which can have an impact on the mechanical properties. The cooling rate should only be fast enough to inhibit crystallization, which can be identified by DSC.

(2)  Melting and crystallization temperatures depend a lot on the molecular weight of the polymer. And any type of polymer synthesis cannot yield just 1 molecular weight, it is always a distribution of molecular weights. So, only stating the average molecular weight is not adequate. There is also the distribution that needs to be considered.

(3)  Data from characterization techniques like XRD, DSC are required to prove that the pellets are either amorphous or crystalline. It should be included in the supporting information. Their reference 35 does provide some information, but the entire DSC curves are not provided in that reference as well. The details in the curves are necessary to understand the whole picture. 

(4)  Following from the previous point, one needs to determine the melting temperature of the polymer (PET) crystals that they are using and then can either use cold-crystallization or melt-crystallization to the crystallize their polymer. Have they found any differences in the adhesion strength when either technique is used to crystallize PET? 

(5)  Polymers typically crystallize as chain-folded lamellae and DSC can be used to determine the thicknesses of these lamellae. XRD can tell what the crystallite sizes are. These quantifications can help get a deeper understanding of the effect of crystallite sizes on the adhesion strength. 

(6)  Further, if the temperature is well below the melting point (meaning severe undercooling), the crystallite sizes would be very small. This will certainly impact their adhesion strength measurements. The method of crystallization, the degree of crystallization, and the size of the crystallites will influence the adhesion strength values which might affect the overall conclusion of which type of statistical distribution the data fall into.

(7)  There are references suggested in this text itself that the statistical analysis of amorphous-crystalline interfaces have already been explored in some detail. For eg: References 35 and 36. A deeper crystallization analysis would improve the paper from the novelty aspect.

(8)  Many of the data points in Figure 2 overlap. So it is hard to appropriately visualize the data. They should modify their presentation methodology and update this figure.  

(9)  What is the significance of this study? How will the conclusion help researchers in polymer science, adhesion science, materials science etc?  The last paragraph in the conclusions section briefly discuss this. However, it would be better if they discuss this in relatively more detail as the audience would appreciate such conclusions and recommendations coming from an expert in statistical analysis as the author themselves. 

Author Response

Responses to Reviewer 2

Many thanks for your evaluation of this work and your detailed comments which are addressed below.

(1)  The processing that has been used to produce amorphous pellets is not well controlled. They say the samples were rapidly quenched from 280 C to ice bath temperature. Such rapid quenching can result in void formation which can have an impact on the mechanical properties. The cooling rate should only be fast enough to inhibit crystallization, which can be identified by DSC.

- There are no other ways to produce the PET samples in the amorphous state except to quench the melt rapidly. Such a drastic drop in temperature, from 280 C to 0 C, was sufficient to actually produce the amorphous PET films, as was confirmed by the DSC curve for such films reported in new reference 42. All the samples were produced at the same conditions and were identical, which is mostly important for the purpose of comparison.

(2)  Melting and crystallization temperatures depend a lot on the molecular weight of the polymer. And any type of polymer synthesis cannot yield just 1 molecular weight, it is always a distribution of molecular weights. So, only stating the average molecular weight is not adequate. There is also the distribution that needs to be considered.

- For a correct comparative study, the most important issue is to use one and the same polymer, i.e. the polymer characterized with identical physical characteristics, including the molecular weight. We have used in our work the PET with one and the same viscosity-average molecular weight 15 kg/mol, i.e. with one and the same set of the chain lengths.

(3)  Data from characterization techniques like XRD, DSC are required to prove that the pellets are either amorphous or crystalline. It should be included in the supporting information. Their reference 35 does provide some information, but the entire DSC curves are not provided in that reference as well. The details in the curves are necessary to understand the whole picture.

- The entire DSC curve confirming the amorphous state of PET used was presented in reference 42 which is now included in the reference list.  

(4)  Following from the previous point, one needs to determine the melting temperature of the polymer (PET) crystals that they are using and then can either use cold-crystallization or melt-crystallization to the crystallize their polymer. Have they found any differences in the adhesion strength when either technique is used to crystallize PET?

- Actually, it would be interesting to compare how the way of crystallization can impact on the PET adhesion behavior, and this issue could be investigated in the future. However, if to melt the cold-crystallized amorphous adhesive joint, its geometry will be changed substantially due to viscous flow. So, the joint will not be appropriate for a correct mechanical testing. In the present work, we have tried to detect the effect of crystallization itself on the adhesion strength distribution. Moreover, the most important was to explore “the jump” from the amorphous state into semi-crystalline one.  

(5)  Polymers typically crystallize as chain-folded lamellae and DSC can be used to determine the thicknesses of these lamellae. XRD can tell what the crystallite sizes are. These quantifications can help get a deeper understanding of the effect of crystallite sizes on the adhesion strength.

- The most important is the same degree of crystallinity after all the crystallization condition employed enabling to make a correct comparison of the adhesion strength distributions.

(6)  Further, if the temperature is well below the melting point (meaning severe undercooling), the crystallite sizes would be very small. This will certainly impact their adhesion strength measurements. The method of crystallization, the degree of crystallization, and the size of the crystallites will influence the adhesion strength values which might affect the overall conclusion of which type of statistical distribution the data fall into.

- The crystallite sizes are not very important for the adhesion strength – the most important is the occurrence of crystallization itself, and it has a negative effect finally arresting the self-bonding process. Once the amorphous joint is crystallized, no further increase in the adhesion strength takes place. The degree of crystallization was identical for all the adhesive joints crystallized at 150 C, resulting in similar and overlapping strength values observed in Figure 2.

(7)  There are references suggested in this text itself that the statistical analysis of amorphous-crystalline interfaces have already been explored in some detail. For eg: References 35 and 36. A deeper crystallization analysis would improve the paper from the novelty aspect.

- The novelty of the work consists in introducing, for the first time, the crystallization factor into the interface self-healing process of the initially amorphous polymer-polymer interface. And the answer to the question raised is received. In the literature, there is a lack of information on this issue.

(8)  Many of the data points in Figure 2 overlap. So it is hard to appropriately visualize the data. They should modify their presentation methodology and update this figure.

- In Figure 2, solid blue symbols for T = 94 C + T = 150 C are now replaced by open ones.  

(9)  What is the significance of this study? How will the conclusion help researchers in polymer science, adhesion science, materials science etc?  The last paragraph in the conclusions section briefly discuss this. However, it would be better if they discuss this in relatively more detail as the audience would appreciate such conclusions and recommendations coming from an expert in statistical analysis as the author themselves.

- In the Conclusion, the following text is added: For a better understanding of the physical processes of various natures (e.g. fracture, adhesion, interface self-healing etc.) in polymers and other materials it is recommended to operate not only with the average value of a property under analysis but with its statistical distribution as well. This approach can give additional useful information and new insight into the molecular mechanisms controlling these processes which can be masked when considering only the average values characterizing them.

Reviewer 3 Report

The paper by Yuri M. Boiko reports a study on self-bonding between pairs of amorphous PET, taken in contact for different times and at different temperatures, with the twofold goal to investigate the best representation of the distribution of adhesion strength as well as to determine the influence of crystallization on initially amorphous structures. The author finds that the Weybull statistics gives a correct description of the adhesion strength distribution and that in the presence of crystallization the diffusion of the interfaces is hindered owing to the formation of crystallites.

The paper is well written, the presentation of the results is clear and the discussion is complete. I propose to publish the paper after the author has taken the following, minor remarks into account:

- lines 118-119: why only T > Tg were chosen? According to the introduction, self-bonding between two contacting polymers occurs also below the glass transition temperature.

- line 119: it should be useful to specify which time intervals were chosen, not only the extreme ones.

- in the caption of figure 2 ‘n’ is defined as the joint number, while in line 134 is the sample number. A unified nomenclature should be preferable.

- in figure 2 it is difficult to follow full symbols with the same colour. Maybe open symbols with different colours would make the figure more readable.

- in figure 4c the root mean square deviation is missing

- why in figure 12 the number of data is 9 instead of 10?

Author Response

Responses to Reviewer 3

Many thanks for your positive evaluation of this work and your comments which are addressed below.

lines 118-119: why only T > Tg were chosen? According to the introduction, self-bonding between two contacting polymers occurs also below the glass transition temperature.

- PET is a crystallisable polymer characterizing with poor self-bonding ability. As is seen in Figure 2, even after contact above Tg, its self-bonding strength is roughly 0.2-0.3 MPa. For comparison, such level of strength for amorphous PS and PPO is achieved at T = Tg-50 C and lower. The corresponding text is added in the last Para of the Introduction (bottom of page 2).

line 119: it should be useful to specify which time intervals were chosen, not only the extreme ones.

-Now, the times used are indicated.

in the caption of figure 2 ‘n’ is defined as the joint number, while in line 134 is the sample number. A unified nomenclature should be preferable.

-Corrected.

in figure 2 it is difficult to follow full symbols with the same colour. Maybe open symbols with different colours would make the figure more readable.

- In Figure 2, solid symbols are replaced by open ones.

in figure 4c the root mean square deviation is missing

- Root mean square deviation is not provided by software in the case of bi-Gaussian fitting.

- why in figure 12 the number of data is 9 instead of 10?

-Because 9 joints were measured for these healing conditions.